# The Changing Landscape of Food Deserts and Swamps over More than a Decade in Flanders, Belgium

**DOI:** 10.3390/ijerph192113854

**Published:** 2022-10-25

**Authors:** Vincent Smets, Jeroen Cant, Stefanie Vandevijvere

**Affiliations:** 1Department of Public Health and Epidemiology, Sciensano [Scientific Institute of Public Health], J.Wytsmanstraat 14, 1050 Brussels, Belgium; 2Research Group for Urban Development, University of Antwerp, Mutsaardstraat 31, 2000 Antwerp, Belgium

**Keywords:** food environment, food deserts, food swamps, supermarket access, modified food environment retail index

## Abstract

Food deserts and swamps have previously been mostly studied in Anglo-Saxon countries such as the USA and Great Britain. This research is one of the first studies to map food deserts and swamps in a mainland European, densely populated but heavily fragmented region such as Flanders. The evolution of food deserts and swamps between 2008 and 2020 was assessed. Special focus was given to areas where high numbers of elderly, young people and/or families with low income live. Food deserts were calculated based on supermarket access within 1000 m and bus stop availability, while food swamps were calculated using the Modified Food Environment Retail Index. The main cause behind the formation of food deserts in Flanders is its rapidly aging population. Food deserts with a higher number of older people increased from 2.5% to 3.1% of the residential area between 2008 and 2020, housing 2.2% and 2.8% of the population, respectively. Although the area that could become a food desert in the future due to these sociospatial and demographic evolutions is large, food deserts are currently a relatively small problem in Flanders in comparison to the widespread existence of food swamps. Unhealthy retailers outnumbered healthy retailers in 74% of residential areas in 2020, housing 88.2% of the population. These food swamps create an environment where unhealthy food choices predominate. Residential areas with a higher number of elderly people, young people and families with low incomes had healthier food environments than Flanders as a whole, because these areas are mostly found in dense urban centers where the ratio of healthy food retailers to all retailers is higher. This research showed that food deserts and swamps could be a growing problem in European regions with a high population density that experience the high pressures of competing land uses.

## 1. Background

For decades, population body weight has been increasing at alarming rates, leading to an obesity pandemic [1]. In Belgium, 55% of the adult population is suffering from overweight and 21% from obesity [2]. Besides a decreased quality of life, individuals with obesity also have higher chances of diabetes, coronary heart disease and several cancers [3,4]. Poor diet quality is considered to be one of the main causes of this obesity epidemic [5]. About 30% of daily caloric intake among both adults and children in Belgium is derived from ultra-processed foods [6]. The food environment has been suspected to be one of the principal drivers of poor diet quality [7,8]. Food environments have commonly been defined as ‘the physical, economic, political and socio-cultural contexts in which people engage with the food system to make their decisions about acquiring, preparing and consuming food’ [1,9]. Today’s food environments exploit people’s biological, psychological, social, and economic vulnerabilities, making it easier for them to eat unhealthy foods’ [10]. 

Some groups are more vulnerable to the detrimental effects of unhealthy food environments than others. For example, families with a poor socioeconomic background, including a low education level, are disproportionately affected [11,12]. In Belgium, the lower educated are 1.5 times more likely to be overweight and twice as likely to be obese [2]. 

The elderly are also strongly affected by their local food environments. As people age, they gradually become less mobile [13]. Car usage declines [14,15] and their average daily walking distance decreases [16]. This restricts the number of food retailers they have access to. The Flemish region is quickly aging. In 2017 20% of the population was over 65, which is expected to rise further to 23% in 2027 and to 25% in 2033. 

The main problem for children and adolescents in regard to the food environment is not the inaccessibility of healthy foods, but the overabundance and marketing of cheap and unhealthy foods [17,18]. In Belgium, 14% of adolescents are overweight [2]. Children and adolescents are more easily influenced by marketing and large food corporations make full use of this to promote their often ultraprocessed products [19,20]. When the number of cheap, unhealthy food options far outweighs the number of healthy options, they will often be tempted to choose the former [21].

The concepts of food deserts and swamps were introduced into the scientific literature to identify neighborhoods with low access to healthy foods. Food deserts have been defined as neighborhoods that lack access to some or all the foods that are required for a balanced, nutritionally adequate diet [22]. Food swamps [23] refer to places where there is an abundance of unhealthy food options relative to healthy food options. The link between food swamps, diet-related behavior and obesity has been clearly established by previous studies. A study in Baltimore, USA, found that adolescent girls living in food swamps consumed more snacks and desserts than girls not living in food swamps [24]. These results were confirmed by [23], who found that food swamps better predict obesity rates than food deserts. The relationship between food deserts, diet and obesity is less clear. Studies found that, although food prices have a larger influence on obesity than the distance to a supermarket [25], better supermarket access can improve dietary behaviors [26]. Similarly, [27] found an association between food deserts and the prevalence of obesity in metropolitan areas, but not in nonmetropolitan areas.

Living in a food desert also increases the risk of cardiovascular disease [28] and decreases the chances of survival for stage II and III colorectal and breast cancer [29]. Food swamps also have been associated with higher hospitalization rates for people with diabetes [30]. Food deserts have been mostly studied in Anglo-Saxon countries such as the USA, the UK, Canada and New Zealand [22]. In the United states, for example, it has been found that communities without adequate access to health affordable food has declined, though more than 5.6 percent of the population still lives in low census tracts [31]. This study found that the likelihood of a census tract being a food desert is strongly correlated with the level of urban sprawl. The authors recommend increasing the land use density, mix and walkability to attract food retailers and mitigate the risk of the census tract becoming a food desert [31]. Overall, while food deserts are quite common in the US, evidence of their existence elsewhere is equivocal [22]. There is significant evidence of the existence of food swamps outside of the US however, e.g., Canada [32] and New Zealand [33].

Recently, research on food deserts outside of the Anglosphere has somewhat increased, especially regarding mainland Europe. A study of Amsterdam did not reveal any issues with access to supermarkets in deprived neighborhoods [34]. A study of Bratislava similarly found access in the study area to be mostly sufficient [35]. Research did find some evidence of food deserts in Nitra (Slovakia), but they were scarce [36]. Neumeier and Kokorsch (2021) [37] identified food deserts in rural Germany, outside of core settlements. In general, however, research on food deserts in mainland Europe remains sparse. Moreover, it mostly focuses on one specific urban area. Research regarding food swamps is even rarer.

Flanders is very different from the USA and most Anglo-Saxon countries. The whole area is densely populated but also heavily fragmented, with small patches of nature-dominated, agricultural- and urbanized land that intertwine [38]. More similar to said countries, urban sprawl has been severe due to lax historical zonal planning regulations. Moreover, Flanders in practice also had very lenient retail zoning laws, allowing significant retail sprawl away from settlements. Due to Flanders’ small surface area and high overall densities, however, it is unclear if this leads to the formation of food deserts. Generally speaking, Flanders is very egalitarian, certainly when compared to the USA [39]. Socioeconomic deprivation might therefore be less of an issue in terms of mobility. Just like most of Western Europe, however, the region is confronted by severe aging, which is likely to be a more important source of immobility [40]. This study will try to answer the question of if the concepts of food deserts and food swamps are applicable to a heavily fragmented and densely populated mainland European, egalitarian but quickly aging region such as Flanders. 

To guarantee equal access to a healthy food environment for every citizen, food deserts and swamps need to be identified. Policies that aim to eradicate diet inequality due to unhealthy food environments can then focus on addressing these vulnerable regions. In Flanders, food deserts [41,42] and swamps [42] have been identified in a few cities on a very local scale. This study is the first to characterize, identify and map food deserts and swamps and their evolution over a period of 12 years for the entire territory of Flanders, Belgium.

## 2. Methods

### 2.1. Study Area

The study area consists of the territory of Flanders, Belgium (Figure 1). Flanders is divided into 300 municipalities and 9194 census tracts. With 6,629,143 people (57.5% of the Belgian population) living on a surface area of 13,625 km^2^, Flanders is strongly urbanized and one of the most densely populated regions in the world. 1,123,000 people (16.9%) in Flanders are of school-going age (2.5 to 17 years) and 1,372,232 are over 65 (20.7%) [43]. 

### 2.2. Data Sources

Several data sets from different sources were used to perform the analysis undertaken in this study (Table 1): 

The commercially available Locatus database [44] was used to map all food retailers in the study area. Locatus is a private company that continuously updates its database through regular field audits. The geographical coordinates, name and address, retail type and floor size of each individual outlet are included in the database. The frequency of field audits varies from once a year—in shopping centers—to once every 2 or 3 years in locations outside of shopping centers. The Locatus database was field validated in the Netherlands and was shown to be highly accurate [45]. It is the only available area-wide retail database in the study region that indexes all individual shops, restaurants and sport/recreation centers selling food and beverages.

For this study, the databases from the years 2008, 2013 and 2020 were acquired. The Locatus database contains 24 types of food retailers that sell food as a primary activity and an additional 25 retailers that sell food as a secondary activity (i.e., including sport and recreation centers). An expert committee in Flanders consisting of 15 dieticians, food scientists and food policy advisors categorized each food retailer type according to healthiness on a Likert scale from 1 to 5 (Appendix A). The inter-rater reliability among experts was high (Gwet AC2 > 0.80) in terms of classifying these types of outlets, meaning there was a high level of agreement among experts on how to categorize these different types of food outlets according to healthiness in the Flemish context. Retailers that scored 4 or 5 were classified as healthy, while retailers that scored 1 or 2 were classified as unhealthy. A Likert score of 3 was considered to be neutral. The retailers that sell food as a secondary activity were all categorized as being unhealthy. Note that these secondary food retailers consist of a wide variety of shops, businesses and entertainment centers. The expert committee concluded that these retailers were most likely to sell unhealthy food and drinks because these types of foods are relatively cheap and addictive, making it more likely for them to be sold and thus generate additional income for the retailers. The Likert scores and subsequent classification in healthy or unhealthy food retailers can be found in additional file 1. Even though supermarkets were classified as neutral by the expert panel, the decision was made to group supermarkets with the healthy retailers in accordance with various studies [23,33,46,47,48]. For 2020, an additional exhaustive dataset including community gardens, farmers’ outlets and farmers’ markets was included in the analysis. All of the latter food retailers were classified as healthy (Likert scale allocation of 5). Because this alternative retail dataset was not available for 2008 and 2013, it was not included in the trend analysis of food deserts and swamps in this study. 

The central reference addresses database (CRAB) database [49] contains all addresses in Flanders. The database is continuously updated and is accessible on the geographical portal of the Flemish government [50]. The enriched database of enterprises (VKBO) [51] contains all addresses of commercial organizations in Flanders. The road network, census tracts and degree of urbanization of Flanders were extracted from the large scale reference database of Flanders (GRB) [52]. A dataset containing all bus stops (situation 2020) was downloaded from the geographical portal of the Flemish government. Income and age-related data on the census tract level were downloaded from national and Flemish statistics services [43,53]. The walkability index, which is a nationwide index that ranks an area according to its relative walkability, was acquired from the website of the Flemish Institute of Healthy Living [54]. 

Population data on the census tract level were stratified by the number of elderly people and young people, the median income decile and walkability score. This data is used to identify food deserts and food swamps in residential areas with the two highest deciles of number of people older than 65 years, younger than 15 years and the two lowest deciles of median income and walkability. Food deserts and food swamps that are calculated based on these deciles will be described as ‘food deserts or food swamps with a high number of elderly, young people, low walkability and/or families with low income’ in the results and discussion section.

#### 2.2.1. Technical Roadmap

The technical roadmap of this study is presented in Figure 2.

#### 2.2.2. Residential Areas

Flanders is a complex mosaic of large urban centers, small towns and villages, industry, agricultural fields and forest patches. This research aims to identify and characterize food deserts and food swamps in residential areas. These residential areas were identified by using the CRAB database. The CRAB database was filtered to only contain residential addresses by subtracting all the commercial addresses contained in the VKBO database. The filtered CRAB database was then combined with the census tracts (n = 9194) of Flanders, the smallest territorial base units in Belgium (average area 1.48 km^2^). The geometric center of gravity of all residential addresses within each census tract was calculated, after which a 1000 m walking distance buffer was created around each geometric center of gravity. The resulting areas were the residential areas in which the food environment was analyzed (Figure 3). When a census tract contained no residential addresses, no residential area was calculated. The 1000 m walking distance buffer was chosen because most people can bridge this distance on foot or by bike in a reasonable time. Other studies, in similar contexts such as [41], assume the same common walking distance. Elderly people however might struggle to walk this distance, certainly with heavy bags of groceries. An extra analysis involving bus stops within a 500 m walking distance from the residential areas was therefore conducted. The resulting residential areas contained 51% of the Flemish territory and approximately 89.5% of all residential addresses, thereby effectively reducing the search for food deserts and swamps to the areas where a large majority of people live. Note that the income- and age-related data used in this study were only available on the census tract level, but because the residential areas housed 89.5% of all residential addresses, this limitation is deemed acceptable. 

The advantage of using geometric residential areas instead of census tracts is that food deserts and swamps can be identified irrespective of administrative boundaries. Census tracts are small territorial units (average size 1.48 km^2^) and habitation is not evenly distributed inside them. People cross the boundaries of these sectors all the time when shopping for groceries without realizing it. The food environment of a Flemish municipality often contains areas of multiple census tracts. Studies that map the food environment on the census tract level [46,55] thus potentially overlook important boundary effects [56,57].

The residential areas of one census tract often cross over into areas of neighboring census tracts, while at the same time not taking into account too much unbuilt land. This results in a more realistic picture of the potential walking behavior of the population and thus provides a solution for the boundary problem. Large urban areas house many people and can be thought of as multiple residential areas with a variety of different food environments. The residential areas often overlap in these urban areas. A certain area can thus be part of multiple residential areas and food environments.

The percentage of the population that lives in these areas was obtained by using the filtered CRAB database to calculate the number of residential addresses in a certain area relative to the number of residential addresses in all the residential areas.

### 2.3. Food Deserts

In Belgium, most people are reliant on supermarkets for their dietary needs. Large supermarkets (>400 m^2^) are dominating total edible grocery sales, with a market share of approximately 85% [58,59]. Within the residential areas, we define a potential food desert as an area where there is no supermarket within a 1000 m walking distance, and no bus stop within a 500 m walking distance where people can take the bus to the supermarket. This more complete definition of food deserts, involving bus stops, was applied to take into account the limited mobility of elderly people, for whom a 1000 m walking distance to and from the supermarket might be too far. 

A 1000 m walking distance buffer around each supermarket in Flanders was calculated to identify the areas in Flanders that lie within a 1000 m of a supermarket. An additional analysis calculated a 500 m walking distance around all bus stops in Flanders. The residential areas that did not overlap with a 1000 m walking buffer from a supermarket or with a 500 m walking buffer from a bus stop were determined next. These non-overlapping areas are the potential food deserts in Flanders. In these potential food deserts, areas with a higher number of older people (>65 years), lower median incomes and lower walkability were identified. The resulting areas were considered the actual food deserts in Flanders. 

### 2.4. Food Swamps

Food swamps were calculated with the Modified Food Environment Retail Index (mFREI) [23,46,47,48]. This often-used metric depicts the ratio of healthy to unhealthy food retailers; it does so by comparing the number of healthy retailers to the number of healthy and unhealthy food retailers. Both retailers that sell food as a primary activity (e.g., fast-food outlets) and retailers that sell food as a secondary activity (e.g., sports centers) were included. The classification of healthy, unhealthy and neutral retailers was performed by an expert committee and can be found in Appendix A. The mFREI is a continuous variable ranging from 0–1. The lower the score, the ‘deeper’ the food swamp is and the more people living in or passing through this food swamp will be tempted to buy unhealthy food. A score of 0 means that an area has no healthy food retailers; no mFREI score was calculated when there were no food retailers in the area. The mFREI was calculated for the residential areas of each census tract. In cases where one or more parts of these areas overlap and the area is part of multiple food environments, the average mFREI score was taken for the overlapping part.

The mFREI was calculated for each residential area with the formula:$$\mathrm{mFREI}=\frac{\#\;\mathrm{healthy}\;\mathrm{retailers}}{\#\;\mathrm{healthy}\;\mathrm{retailers}+\#\;\mathrm{unhealthy}\;\mathrm{retailers}}$$

The results were rasterized to a resolution of 100 × 100 m, and subsequently the mFREI scores are discretized to a precision of 0.1. Additionally, a stratification of the mFREI score was determined according to the level of urbanization.

## 3. Results

### 3.1. Food Deserts

The evolution of food deserts in Flanders in the identified residential areas between 2008 and 2020 is shown in Table 2. The total potential food desert area stayed relatively constant between 2008 and 2020, covering 56.3% and 56.9% of the residential areas, respectively, and housing 27.4% of the population in 2008 and 28.3% of the population in 2020. The area considered a food desert with a high percentage of people >65 years slightly increased between 2008 and 2020 from 2.5% of the residential areas in 2008 to 3.1% in 2020. The percentage of the population that lives in these food deserts increased from 2.2% to 2.8% of people that live in residential areas. In contrast, the food deserts area where low-income families are overrepresented decreased from 7.8% of the residential areas in 2008 to 4.7% in 2020. The percentage of the population that lives in these deserts decreased from 1.7% in 2008 to 1.1% in 2020. No data were available for food deserts with low walkability before 2020. The food deserts area with lower walkability was similar in size to that of the food desert area where low-income families are overrepresented and covered 4.9% the residential areas in 2020, housing 1% of the population. In total, 11.8% percent or 811.5 km^2^ of the residential areas were considered to be a food desert in 2020, and were inhabited by 4.9% of the population (Figure 4). Note that the sum of all three types of food deserts (>65 years, low income and walkability) is larger than the total food desert area because of overlaps between the different types of food deserts.

### 3.2. Food Swamps

Between 2008 and 2020, the mFREI scores in the residential areas have gradually decreased. Figure 5 and Figure 6 show the evolution of food swamps on an area and population basis, respectively. 

The trend analysis in this paragraph will be for the years 2008, 2013 and 2020a. The next paragraph will compare the year 2020 with Locatus data (2020a) on the one hand, and with Locatus data and the extra dataset of community gardens, farmers’ markets and farmers’ outlets on the other hand (2020b). A significant portion of the residential areas contained no stores at all (Figure 5). These areas are sparsely populated (Figure 6). In 2008, 8.8% of the population lived on 23.2% of the residential area that contained no food stores. In 2020, this number has decreased slightly, with 8.2% of the population living on 21.6% of residential area without food stores. A large part of the residential areas only contains unhealthy food stores (mFREI = 0). The residential area that exclusively contains unhealthy food stores increased from 28.7% in 2008 to 33.8% in 2020 and is inhabited by 19.8% and 24% of the population, respectively. In general, unhealthy food retailers outnumber healthy food retailers in the majority of residential areas (mFREI < 0.5). In 2008, 71% of the residential area had an mFREI score under 0.5. This area contained 86% of the population. In 2020 this number had increased to 74% of residential area, housing 88.2% of the population. In contrast, the residential areas where healthy food retailers outnumber the unhealthy ones are very small. The percentage residential area with an mFREI > 0.5 was 1.2% and 2.1% in 2008 and 2020, respectively, housing 2.1% and 1.6% of the population.

When comparing the year 2020, to results with (2020b) and without (2020a) the dataset of community gardens, farmers’ markets and farmers’ outlets, a strong difference is seen both in Figure 5 and Figure 6. Because community gardens, farmer’s markets and farmers’ outlets were all classified as healthy (Likert scale allocation 5), both the residential area and the percentage of the population living in that area with low mFREI scores (<0.5) decreased, while the residential area and percentage of the population living in areas with high mFREI scores (>0.5) increased significantly. For example: when including the dataset of community gardens, famer’s markets and farmers’ outlets, healthy retailers outnumbered unhealthy retailers (mFREI >0.5) on 8.4% of the area, with 5.2% of the population living in that area. Without the extra dataset, this number drops to 2.1% of the area and 1.6% of the population.

Table 3 and Figure 7 show the percentages of residential areas with a higher number of people older than 65 years, younger than 15 years, families with low income and low walkability for the year 2020. These scores are calculated for the dataset that includes the community gardens, farmers’ markets and farmers’ outlets.

Although the majority of the population lives in areas with low mFREI scores, some differences are found between the scores of areas with a higher number of persons older than 65 years, younger than 15 years, families with low income and low walkability on the one hand (Figure 7), and the scores for all residential areas of Flanders on the other hand (Figure 7, 2020b). For higher number of persons older than 65 years, younger than 15 years and for low-income families, the proportion of the population that lives in areas without an mFREI score or an mFREI score of 0 is substantially smaller than that of all residential areas of Flanders. The proportion of the population that lives in these subareas (older, younger, low income) with an mFREI score of 0.1 is similar as the residential areas average in Flanders but is larger for the mFREI scores 0.2 and 0.3. From mFREI score 0.4 onwards, the proportion of the population that lives in these areas is relatively similar. Interestingly, both the area where a lot of older people and a lot of younger people live consists of around one fifth of all residential areas, and around 40% of the population resides there. The areas with low income consist of only 13.7% of the area, and around 20% of the population lives in this area. 

The food swamps with a low walkability differ strongly from the previous food swamps discussed. Over half of the population that lives in residential areas with low walkability have no food retailers nearby. Another 24.6% live in areas with an mFREI score of 0.1. A significant proportion (12.8%) of the population lives in areas where only healthy food retailers are present. This is likely because of the inclusion of the dataset of farmers’ shops and community gardens, which are more often than traditional retailers located in rural, remote areas. The total residential area with low walkability is small however, consisting of only 5.2% of the total residential area, containing only 1.3% of the population (Table 3).

The geographical distribution of food swamps in 2020 is shown in Figure 8 and the mFREI scores according to the level of urbanization are shown in Figure 9.

The variation in mFREI score increases from urbanized regions to rural regions. More urbanized regions have the highest median mFREI score because they have a higher density of healthy compared to unhealthy food retailers than other areas (Figure 9). The periurban areas have less healthy food retailers, resulting in a lower median mFREI score and a wider variation. The more rural areas in Flanders have the lowest mFREI scores and display the widest variation.

## 4. Discussion

### 4.1. Food Deserts

In terms of the number of inhabitants, food deserts are a smaller but not negligible problem than food swamps in Flanders. In 2020, 11.8% of the Flemish residential areas, housing 4.9% of the population, could be considered a food desert (Table 2). These results are backed by previous research that found that the problem of food deserts in Flanders to be highly localized and contextualized [41]. Despite there being strong differences in land use and population density, these findings are comparable to results from research carried out in other countries. Research conducted in the USA found that a similar percentage (5.6%) of the population still lives in census tracts with low access to food retailers [60]. There are however strong regional differences in food deserts in the USA, and food deserts there tend to disproportionately affect mostly poor and particularly African American neighborhoods. Based on our results, the food deserts with a higher number of people >65 years are the most concerning as they grew in area between 2008 and 2020 (2.5% to 3.1% of residential area), and an increasing number of people live in them (2.2% to 2.8% of the population). The food deserts in regions with many low-income families have decreased with 212.4 km^2^ between 2008 and 2020 and now only house 1.1% of the population in Flanders. Because of the more egalitarian society in Flanders in comparison to other countries such as the USA, food deserts due to socioeconomic causes are less of an issue. A general picture of food desert evolution between 2008 and 2020 is not available because walkability data were not available for 2008 and 2013. Despite the fact that the actual food deserts in Flanders are relatively small in size and population, it is important to note that the potential food deserts in Flanders cover a large area. In 2020, 56.9% of the residential area was a potential food desert, housing 28.3% of the population. These potential food deserts are at risk of becoming actual food deserts, as the population in Flanders is quickly aging [61]. This is already happening, as the food deserts with many people older than 65 years increased in area and population between 2008 and 2020. Precautionary policies to avoid these potential food deserts becoming actual food deserts are therefore advisable. 

### 4.2. Food Swamps

Food swamps are a far more significant problem in Flanders. The residential areas where unhealthy retailers outnumber healthy retailers (mFREI score 0 to 0.4) increased slightly between 2008 and 2020 from 71% to 74%, housing 86% and 88.2% of the population living in residential areas, respectively. In 2020, 25.2% of the residential areas was found to contain only unhealthy food retailers, housing 17.2% of the population, even when including an additional dataset containing community gardens, farmers’ outlets and farmers’ markets. Residential areas with substantial numbers of elderly people, younger people and people with low incomes are less unhealthy when compared to Flanders as a whole, with larger parts of the population living in areas with ‘intermediate’ mFREI scores of 0.2 and 0.3. 

This is in contrast to international findings that found food swamps were disproportionately located in disadvantaged areas [33,62]. The reason that these findings in Flanders are different is that most elderly people, young people and families with low incomes reside in urban centers, where there are higher densities of healthy food stores as compared to suburban and rural areas in Flanders [41]. Even though these central urban areas can be considered less unhealthy, the number of healthy food retailers are still far outnumbered by the number of unhealthy food retailers, meaning that unhealthy snacks or meals are still very easy to find in comparison to healthy snacks or meals. Diet inequality is closely interwoven with the history and culture of a country and with its historical spatial planning policies. A careful analysis of each specific country and region should therefore be made when assessing diet inequality, without generalizing or extrapolating findings from one region to another.

### 4.3. Strengths and Limitations

This is the first study that analyzes the food environment in the whole of Flanders over a 12-year time period. One of the strengths of this study is that it provides a complete picture of the food environment on the neighborhood level. Food deserts and swamps were identified in residential areas for vulnerable population groups. The calculated residential areas are a more natural representation of the food environment than the census tracts used in other studies [56,57]. Another strength of this study is the use of the Locatus database, which is the most exhaustive retail database available in Flanders. Both retailers that sell food as a primary and secondary activity are included in the analyses, giving a complete picture of the food environment.

A limitation of this study relates to the calculation of the mFREI. The classification of healthy, unhealthy and neutral food retailers remains somewhat subjective. In some cases, e.g., fast-food restaurants, the classification as unhealthy is obvious. In other cases, such as supermarkets, the classification is harder. Given that the edible groceries retail landscape in Flanders is dominated by supermarkets, this classification is important for our results. Most food swamp studies classify supermarkets as healthy [23,33,46,47,48] and this study chose to do the same, even though the expert panel classified supermarkets as ‘neutral’ with a Likert score of 3. The decision to include supermarkets among the healthy food retailers was taken because, even though many people will buy unhealthy food at a supermarket, many rely on supermarkets as their primary retailer for healthy products. Moreover, most large supermarket chains in Flanders have made commitments to creating healthier in-store environments via methods such as, for example, promoting more products with a healthy nutrition label. Other food retailers such as fast-food outlets and night shops predominantly market and offer unhealthy food. People nearly always visit these stores with the intention of buying unhealthy, processed food. The number and density of these types of food retailers saw significant increases in the last decade [63,64]. The increasing obesity rates observed in western countries are likely to be at least in part because of this increased abundance of cheap, heavily marketed and easy to come by unhealthy food products offered by these retailers. 

Another possible limitation of this study is that the frequency of visits was not taken into account for different food retailers. People will buy more food from retailers that sell food as a primary activity such as fast-food restaurants than retailers that sell food as a secondary activity such as theaters. The concept of food swamps however represents the availability of unhealthy food in comparison to healthy food. By this measure, the frequency of visit does not matter, but the widespread availability does. Even though the retailers that sell food as a secondary activity do not make up for a large portion of the sale of unhealthy food, they do help to create an unhealthy food environment by exposing people to unhealthy food even when engaging in other activities. 

A walking distance of 1000 m was used to delineate the residential areas, which serve as the borders of the food environment. For some very active and mobile people, the food environment that was calculated in this study will be too small. They will easily walk, bike or drive further than 1 km to buy food. For other, often elderly and other less mobile people, the food environments as defined in this study will be too big. People that live near the edge of the calculated residential areas could travel outside the boundary to buy food, although it is more likely that they will travel in the direction of the mass center of their area of residential settlement because most food retailers are located in villages- and towns’ centers [65]. This study does not try to characterize the food environment for individual persons on a very local scale but aims to give an overview of the evolution of the food environment since 2008 on a regional scale. This inevitably results in a trade-off where detail is sacrificed at the expense of scale. 

### 4.4. Policy Recommendations and Future Research

The results of this research can be used to inform policy makers about the situation of their local food environment. Actual food deserts and swamps can be mitigated through the establishment of more healthy food retailers in these areas. In the case of food swamps, efforts can be made to increase the number of healthy food retailers but also to limit the number of unhealthy food retailers. A legislative framework needs be developed to efficiently address these issues. Policymakers should also take special note of the potential food deserts in Flanders which compromise over a half of the Flemish residential area, especially those where many middle-aged persons live. With the quickly aging Flemish population these areas are in danger of becoming actual food deserts in the near future.

Further research should focus on the integration of the food environment on the local scale and the regional scale. More research that links the health of the food environment with the weight status of the people that live there also needs to be performed.

## 5. Conclusions

This study is one of the first studies to map the healthiness and evolution of retail food environments in a densely populated and fragmented area such as Flanders by using the concepts of food deserts and food swamps. The results show that socioeconomic inequalities are less of an issue with regard to the existence of food deserts compared to other, less egalitarian countries such as the USA. The main source of immobility and formation of food deserts in Flanders is its quickly aging population. The results show that, while food deserts with a higher number of low-income families decreased between 2008 and 2020, the food deserts with a higher number of people older than 65 years increased. Considering Flanders’ rapidly aging population, the area that could potentially become a food desert is large, compromising over half of the residential area in Flanders. Food swamps were ubiquitous in Flanders. The food swamp area and population was already high in 2008 and further increased in 2013 and 2020. Areas with a higher number of older people, young people and families with low incomes were healthier than Flanders as a whole, because these areas are mostly found in dense urban centers where the ratio of healthy food retailers to all retailers is higher. The widespread occurrence of food swamps may exacerbate the obesity epidemic in Flanders. This research shows that food deserts and swamps are not only an issue in the Anglosphere but could also be a growing issue in a small, dense, highly urbanized, mainland European region experiencing high land use pressures. Policymakers can use these findings to create healthier food environments to combat the ongoing obesity epidemic.

## Figures and Tables

**Figure 1 ijerph-19-13854-f001:**
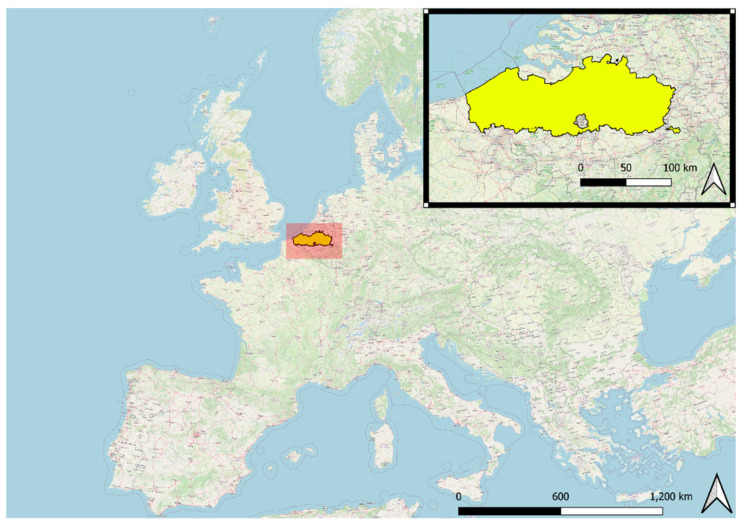
The location of the study area in Europe.

**Figure 2 ijerph-19-13854-f002:**
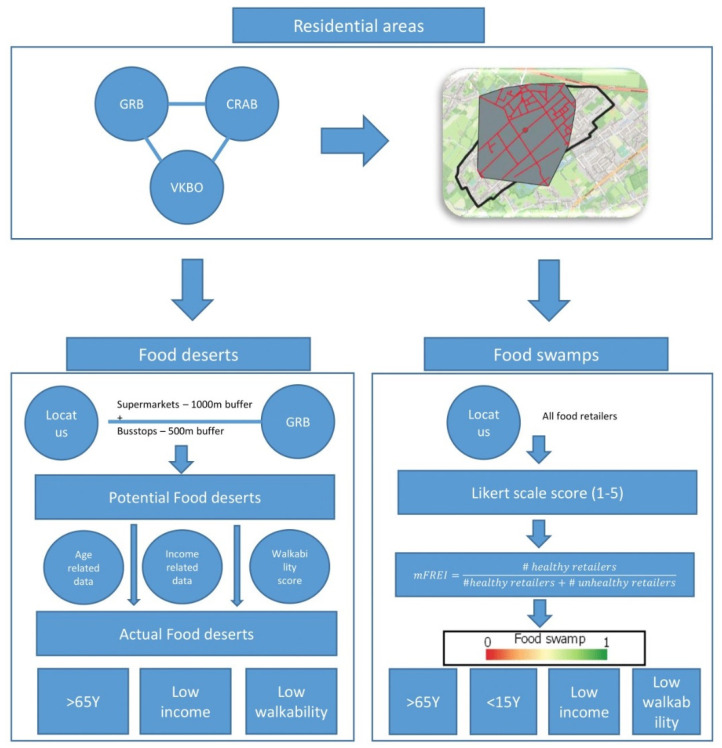
Technical roadmap of the research. Firstly, the residential areas in Flanders are identified. Secondly, the food deserts and food swamps are determined within these residential areas.

**Figure 3 ijerph-19-13854-f003:**
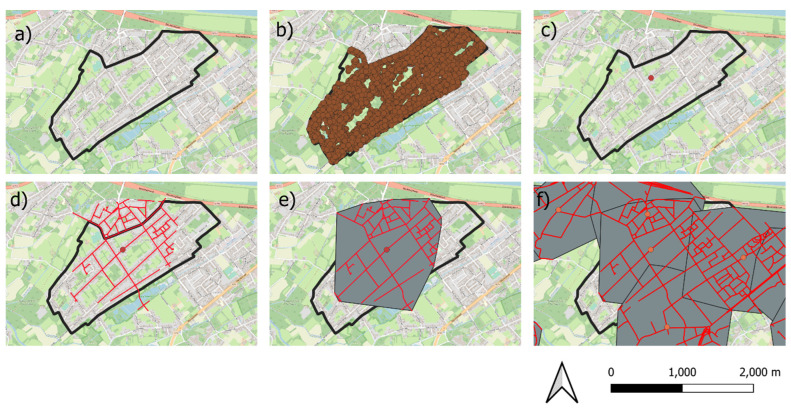
Procedure to create the residential areas. A census tract (**a**) was filtered to only contain the points of the modified CRAB database, depicting the residential addresses (**b**). The geometric center of gravity was calculated for these points (**c**). Next, the 1000 m road distance from the geometric center of gravity was calculated (**d**). A buffer containing the 1000 m road distance measurement was created, resulting in the residential areas (**e**). This procedure was carried out for each census tract in Flanders with residential buildings (**f**). Note that the residential areas often overlap.

**Figure 4 ijerph-19-13854-f004:**
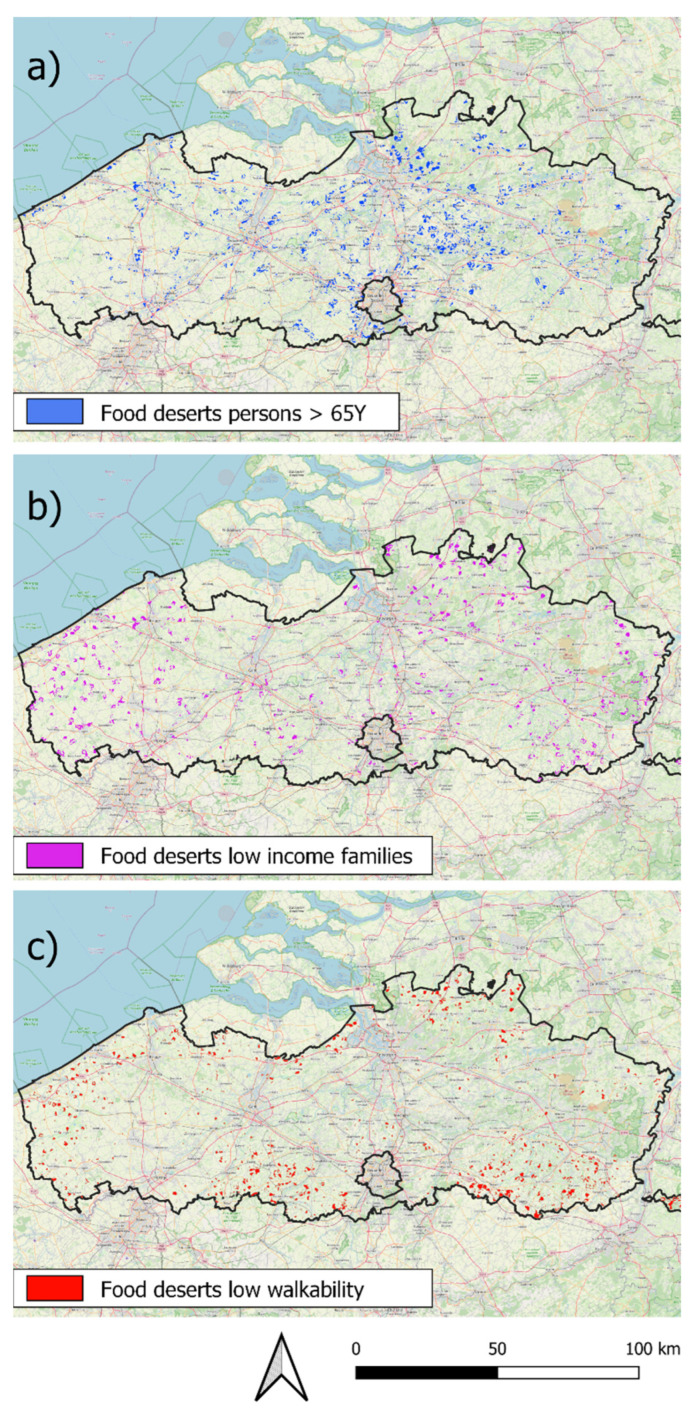
The geographical dispersion of food deserts in Flanders (data 2020). (**a**) Food deserts in areas with a high number of people >65Y; (**b**) food deserts in areas with a high number of families with low income; (**c**) food deserts in areas with low walkability.

**Figure 5 ijerph-19-13854-f005:**
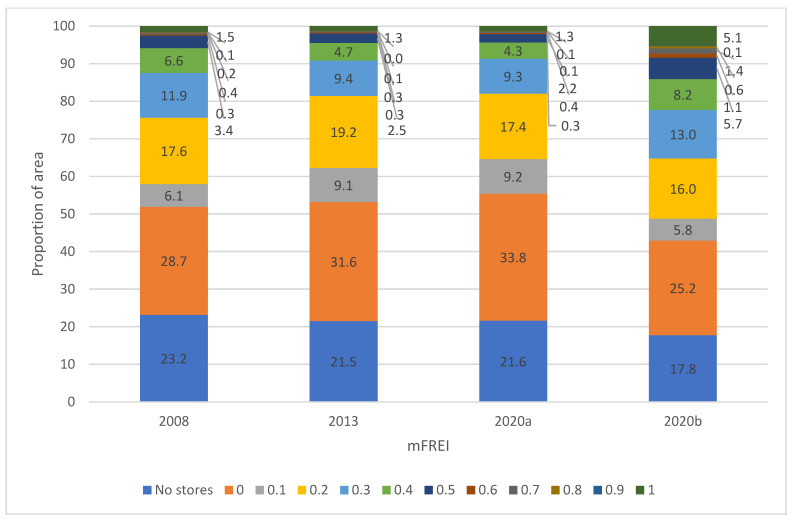
The evolution of food swamps in residential areas between 2008 and 2020, expressed on an area basis. The proportions are based on the total area of the residential areas. 2008, 2013 and 2020a are the food swamp scores based on Locatus data only. 2020b are the food swamp scores based on the Locatus data and the extra dataset of community gardens, farmers’ markets and farmers’ outlets.

**Figure 6 ijerph-19-13854-f006:**
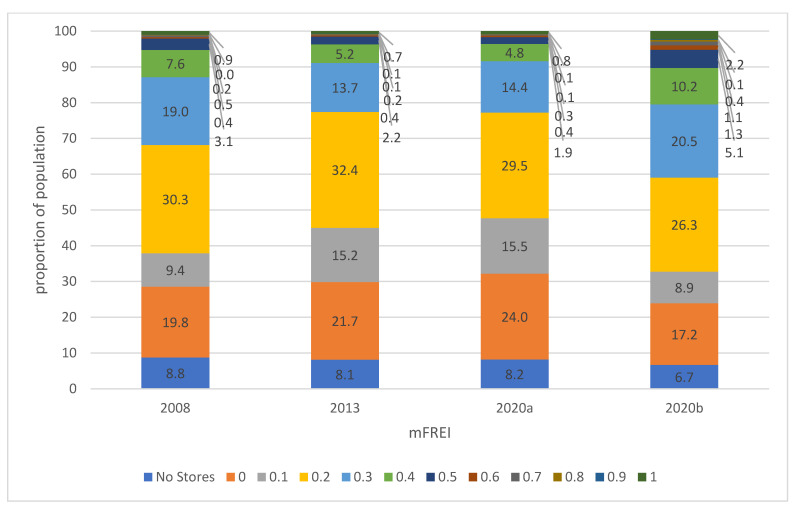
The evolution of food swamps in residential areas between 2008 and 2020, expressed on a population basis. The proportions are based on the total number of people living in the residential areas. 2008, 2013 and 2020a are the food swamp scores based on Locatus data only. 2020b are the food swamp scores based on the Locatus data and the extra dataset of community gardens, farmers’ markets and farmers’ outlets.

**Figure 7 ijerph-19-13854-f007:**
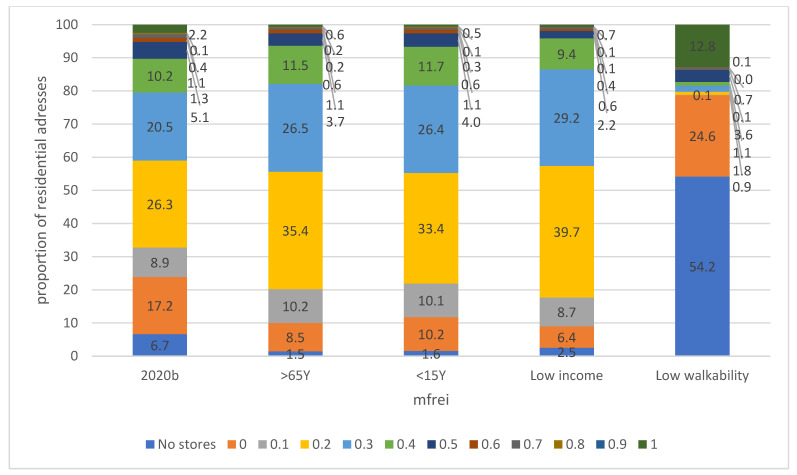
Food swamps in residential areas with a lot of people > 65 years and younger than 15 years; for people with low incomes and regions with low walkability for the year 2020, data includes farmers’ shops and community gardens. 2020b is identical, as in Figure 6, and is added for comparison purposes. The mFREI scores are expressed on a population basis.

**Figure 8 ijerph-19-13854-f008:**
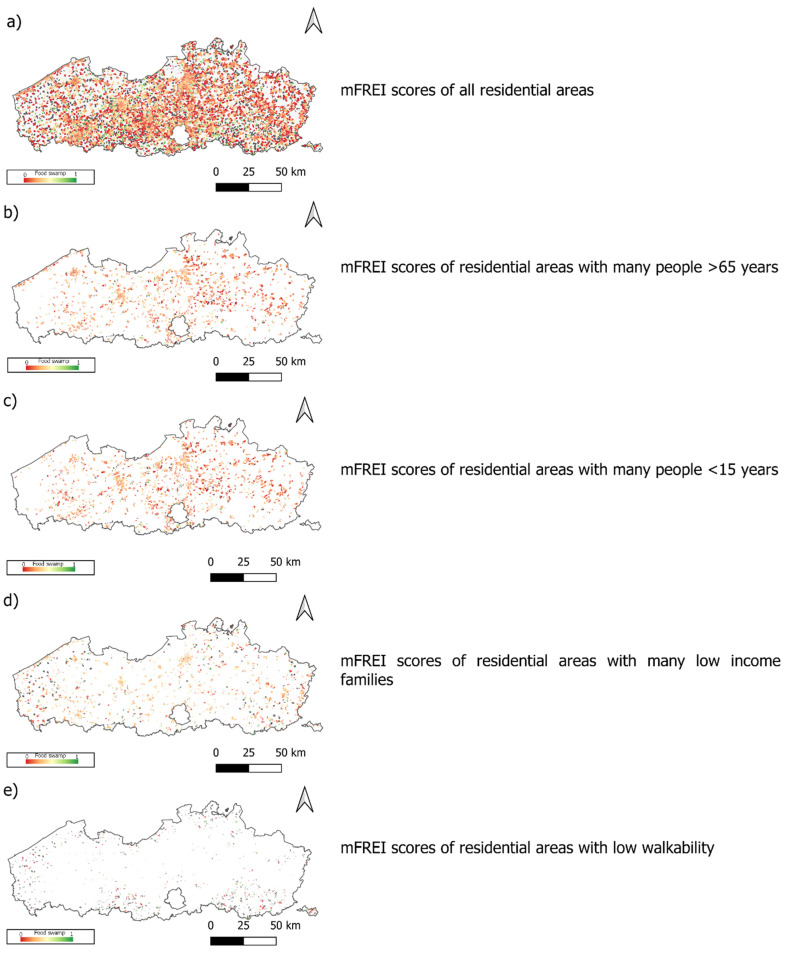
(**a**) The geographical dispersion of food swamps (mFREI scores) in Flanders; (**b**) food swamps in areas with a high number of people > 65 years; (**c**) food swamps in areas with a high number of people < 15 years; (**d**) food swamps in areas with low incomes; (**e**) food swamps in areas with low walkability (data from 2020 and includes community gardens, farmers’ markets and farmers’ outlets). The scale goes from mFREI score ‘0′ or very unhealthy (dark red) to mFREI score ‘1′ or very healthy (dark green). The dark grey areas are residential areas without any food retailers.

**Figure 9 ijerph-19-13854-f009:**
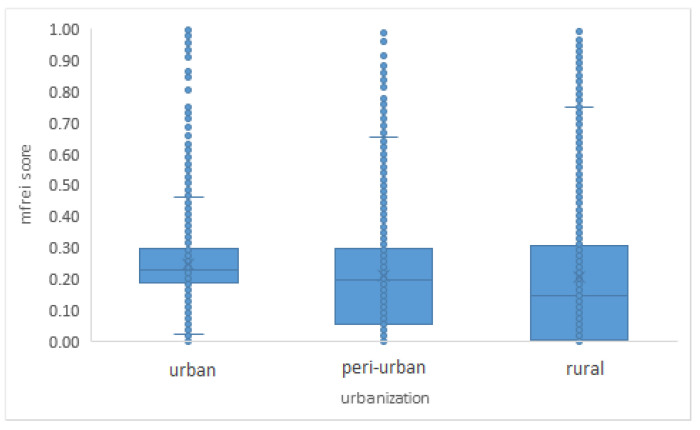
mFREI scores according to level of urbanization. The data is from 2020 and includes community gardens, farmers’ markets and farmers’ outlets.

**Table 1 ijerph-19-13854-t001:** The different datasets used in this study.

Dataset	Source	Description
Locatus	Locatus company	Dataset containing all food retailers and restaurants in Flanders for the years 2008, 2013 and 2020.
GRB	Flemish government	Dataset containing all roads, census tract boundaries, bus stops and the degree of urbanization in Flanders.
CRAB	Flemish government	Central reference addresses database. Contains all official addresses in Flanders.
VKBO	Flemish government	Dataset containing all addresses of public and private companies in Flanders.
Income related data	National and Flemish statistical services	Dataset containing income related data on the census tract level.
Age related data	National and Flemish statistical services	Dataset containing age related data on the census tract level.
Walkability score	Flemish institute of healthy living	Dataset that scores each area based on its walkability.

**Table 2 ijerph-19-13854-t002:** Potential food deserts and actual food deserts in Flanders for people >65 years, low incomes families and with low walkability. The percentages are expressed relative to the residential areas. No time comparison can be performed for food desert low walkability and total food desert area because walkability data are only available for 2020 onwards.

Indicator	2008	2013	2020
Potential food desert [km^2^]	3903.9	3873	3956.6
Potential food desert [% area]	56.3	55.7	56.9
Potential food desert [% population]	27.4	26.9	28.3
Food desert >65Y [km^2^]	180.2	191.7	218.7
Food desert >65Y [% area]	2.5	2.7	3.1
Food desert >65Y [% population]	2.2	2.3	2.8
Food desert low incomes [km^2^]	546.2	408.1	333.8
Food desert low incomes [% area]	7.8	5.9	4.7
Food desert low incomes [% population]	1.7	1.2	1.1
Food desert low walkability [km^2^]	n.a.	n.a.	340.2
Food desert low walkability [% area]	n.a.	n.a.	4.9
Food desert low walkability [% population]	n.a.	n.a.	1.0
Food desert total [km^2^]	712.6 *	583.0 *	811.5
Food desert total [% area]	10.2 *	8.4 *	11.8
Food desert total [% population]	3.8 *	3.7 *	4.9

* Food desert total area and percentage of 2008 and 2013 are calculated without walkability data and cannot be compared with food desert total area of 2020.

**Table 3 ijerph-19-13854-t003:** The percentage of the residential area and percentage of the population in residential areas that lives in areas with a lot of older people, a lot of younger people, low incomes or low walkability. The percentages are based on the mFREI scores that are calculated with the Locatus data and the extra dataset of community gardens, farmers’ markets and farmers’ outlets.

	>65Y	<15Y	Low Income	Low Walkability
% area	19.1%	19.6%	13.7%	5.2%
% population	42.9%	44.0%	20.0%	1.3%

## Data Availability

The datasets used and/or analyzed during the current study are available from the corresponding author on reasonable request.

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
