# Peer review of "The Changing Landscape of Food Deserts and Swamps over More than a Decade in Flanders, Belgium"

_ijerph, 2022, doi:10.3390/ijerph192113854_

Round 1
Reviewer 1 Report
This is a well-established and informative study of food deserts in Belgium. However, I would like to recommend some revisions to improve the quality and the contribution of the study.
The Abstract section should be trimmed and the names of the subsections, such as methods and others, should be removed. As far as I understand, the IJERPH format requires not dividing the abstract into sections. The author should make the narrative as brief and informative as possible - overall relevance of the topic, relevance for the particular territory, knowledge/practice gaps, methods to address the gaps, major findings (no details, such as percentages, are needed here), and implications.
In the Introduction, the relevance of the study should be better explained. For a common reader, the need for the identification of food deserts throughout Belgium may be unclear. The author mainly refers to the USA, but the discussion of food deserts problem in EU countries would be more helpful for this particular study. The exact relevance of the study for Belgium is poorly articulated.
Reviewer 2 Report
This manuscript (ijerph-1945535) tries to analyze the changing landscape of food deserts and food swamps from 2008 to 2013, and to 2020 in Flanders, Belgium. Overall, it is a well-written and scientifically sound research. Nevertheless, several major issues still need to be carefully solved before it can be further considered. More detailed comments and suggestions are shown as follows:
-1. In the Introduction Section, the authors have devoted too much space to describing the background of the food problems in the study area (Flanders, Belgium), which should be briefly mentioned in one simple subsection or paragraph. In addition, the authors also need to look further into the latest research in this field. In fact, the literature review is far from enough. Therefore, a thorough and criticism-featured Literature Review Section is needed.
-2. In the Data sources Section, why not use the well-accepted points of interest (POI) data with more abundant and detailed information?
-3. An expert committee in Flanders consisting of 15 dieticians, food scientists and food policy advisors categorized each food retailer type according to healthiness on a Likert scale from 1 to 5. This determination method is a bit subjective. Are there any more objective and quantitative methods?
-4. Please explain clearly why the retailers that sell food as a secondary activity were all categorized as unhealthy.
-5. Even though supermarkets were classified as neutral by the expert panel, the decision was made to group supermarkets with the healthy retailers in accordance. This operation sounds a bit weird and did not respect the opinions from the expert panels.
-6. I suggest the authors to provide the location map of the study area. Besides, a technical roadmap of this research is also helpful and necessary.
-7. In addition, the authors should better summarize and illustrate all the datasets used in this study in a new table and a figure, respectively.
-8. The advantage of using geometric residential areas instead of census tracts is that food deserts and swamps can be identified irrespective of administrative boundaries. However, the income and age information were summarized at the administrative level.
-9. Only the geographical dispersion of food deserts in Flanders in 2020 was presented. I suggest the authors to include the data in 2008 and 2013. The same suggestion for the food swamps dispersion.
-10. Please clearly explain the differences of your result and the results from "Food inaccessibility in Flanders identifying spatial mismatches between retail and residential patterns".
-11. The Discussion and Conclusion Sections failed to engage with the wider readership of this international journal. For example, most contents in the abstract are only related to the specific study area. The novelty and originality should be clearly justified that the manuscript contains sufficient contributions to the new body of knowledge from the international perspective.
-12. The format of citation in text is incorrect.
Round 2
Reviewer 2 Report
The authors responded adequately to my comments and suggestions.